# Fine-Spatial Boreal–Alpine Single-Tree Albedo Measured by UAV: Experiences and Challenges

**Eirik Næsset Ramtvedt ***, **Terje Gobakken** and **Erik Næsset**

Faculty of Environmental Sciences and Natural Resource Management, Norwegian University of Life Sciences, P.O. Box 5003, NO-1432 Ås, Norway; terje.gobakken@nmbu.no (T.G.); erik.naesset@nmbu.no (E.N.)
* Correspondence: eirik.nasset.ramtvedt@nmbu.no

**Abstract:** The boreal–alpine treeline is a fine-spatial heterogeneous ecotone with small single trees, tree clusters and open snow surfaces during wintertime. Due to climate change and decreased grazing of domestic animals, the treelines expand both upwards into the mountains and northwards into the tundra. To quantify and understand the biophysical radiative climatic feedback effect due to this expansion, it is necessary to establish measurement strategies of fine-spatial albedo by which relationships with the tree structure and snow-masking effect can be quantified. In this study, we measured single-tree Norway spruce albedo for small trees using an unmanned aerial vehicle (UAV). The platform allows the measurement of proximal remotely sensed albedo, enabling the provision of fine-spatial reflectance distributed over larger geographical areas. The albedo measurements varied from 0.39 to 0.99. The interaction between the diurnal course of the sun and sloping terrain constituted the most important driving factor on the albedo. Surprisingly, all tree structural variables revealed smaller correlations with the albedo than typically found for boreal and boreal–alpine summertime albedo. The snow-masking effect of the trees on the albedo was statistically significant and was found to be stronger than the effects of tree structural variables. Only the canopy density had a statistically significant effect on the albedo among the tree structural variables. This was likely explained by the imprecise heading of the hoovering positions of the UAV and insufficient spatial resolution of the reflected radiation measurements. For further development of fine-spatial UAV-measured albedo, we recommend the use of UAVs with high-precision navigation systems and field-stop devices to limit the spatial size of the measured reflected radiation.

**Keywords:** albedo; fine-spatial; Norway spruce; proximal remote sensing; small trees; unmanned aerial vehicle

## 1. Introduction

Global warming is leading to rapid shifts of trees and shrubs in boreal–alpine, sub-arctic and boreal-tundra ecotones [1–4]. Since the trees grow at their temperature tolerance limit in these areas, even a moderate increase in temperature may lead to the migration of treelines and colonization of treeless areas [5], as well as rapid increases in the growth of existing trees [6,7]. Additionally, the change in herbivory has been shown to be an important reason for changing treelines [8–10]. Land abandonment and decreased grazing of domestic animals in montane areas since the mid-19th century have caused the treelines to shift upwards [11], both in terms of higher elevations and latitudes. The boreal forest and boreal–alpine treeline have a warming effect on the climate by reducing the surface albedo [12,13]. This is particularly pronounced in wintertime due to the large differences in reflectance between the trees and snow surfaces. As result, treeline migration and forest expansion into alpine, tundra and sub-arctic regions in Scandinavia have been found to play critical roles in climate warming in wintertime by reducing surface albedo [14,15].

Because of the large spectral contrast between snow-covered surfaces and snow-free trees, the wintertime forest albedo is mainly dependent on the composition and

distribution of trees and open ground. The differences in albedo can be substantial [16,17], typically ranging from 0.07 for bare canopies of Norway spruce [18] to 0.98 for non-polluted snow [19]. However, the effective forest snow albedo is also affected by the process of multiple reflection and scattering [20,21], being dependent on the three-dimensional structure of the tree. When the radiation travels through the canopy, different regimes of snow masking and canopy shading occur. Especially for large solar zenith angles, the direction of the incoming solar radiation interacts with the tree canopy and creates shaded and sun-lit areas on the snow surface, which have different albedo under the conditions of above-canopy incoming solar radiation, remaining spatially consistent [22]. Differences in canopy density and the degree of snow interception, therefore, impact the effective forest snow albedo. It has also been widely shown how tree structures and forest characteristics affect wintertime and even summertime boreal albedo. Albedo decrease with increasing tree height [22,23], increasing leaf area index (LAI) [24], increasing biomass [25,26], increasing growing stock volume [23], increasing canopy cover [22,23,26,27], increasing age [27–29] and increasing/decreasing fractions of coniferous/deciduous species [23,25–27,30].

Boreal–alpine treelines are highly heterogeneous and consist of both scattered single trees and denser and more continuous tree canopies. To capture the fine-spatial variations between highly reflective snow and darker trees, albedo measurements with a spatial resolution corresponding to the size of the objects of interest is paramount. Currently, there are no existing sources that regular provide albedo data at this spatial resolution. Traditional satellite products are offered at spatial scales that are too coarse for accurate albedo representation of fragmented vegetation [31] and fail to capture the radiative effects between open surfaces and denser canopies. Even fixed-station radiation measurements usually have a spatial resolution that is too coarse for such purposes; neither are they capable of providing albedo data over larger spatial extents. Easily movable telescopic radiation towers have been used to provide single-tree summertime mountain birch albedo in the boreal–alpine treeline [32]. However, this type of in situ measurement is time consuming and limits the maximum tree height that can be measured, meaning it is not suitable for taller canopies. Recent studies have successfully demonstrated the use of an unmanned aerial vehicle (UAV) to provide meaningful albedo observations with both high spatial and temporal resolutions [22,33,34]. As an example, the UAV-measured albedo of a discontinuous fine-scaled heterogeneous Norway spruce forest was related to the canopy cover and tree height during different meteorological conditions [22]. However, the hoovering heights above the top of canopy in this study were 10–15 m, corresponding to footprints of reflected radiation of 1.25–2.82 ha.

The main objective of the present study was to provide single-tree wintertime measurements of albedo in the boreal–alpine treeline by using an UAV and to gain knowledge for the further development of measurement strategies of fine-spatial proximal remotely sensed albedo for heterogeneous environments. Specifically, the study aimed to determine the importance in controlling Norway spruce single-tree boreal–alpine albedo based on the three following factors: (1) the tree snow-masking effect; (2) the effects of different tree structural variables; (3) the effect of the diurnal solar variation. Seven repeated UAV flights were performed for 15 Norway spruce tree objects on a mountain ridge in southern Norway during the period of the 15–19 April 2021.

## 2. Materials and Methods

### 2.1. Study Site Description and Ground Measurements

The study site (60°'N 9°01'E) is located in the treeline on a mountain ridge extending in the north–south direction in the municipality of Rollag, southern Norway (Figure 1a). The main tree species in the study area are Norway spruce (*Picea abies* (L.) Karst), mountain birch (*Betula pubescens* ssp. *czerepanovii*) and Scots pine (*Pinus sylvestris* L.). Fifteen tree objects (Figure 1b) were selected subjectively to cover a range in size, structure and terrain slope and aspect during field work on the 28th of June 2020. The tree objects were either

single trees or small tree clusters of Norway spruce (see illustrations in Figure 3). The elevation ranges from 910 to 950 m above sea level.

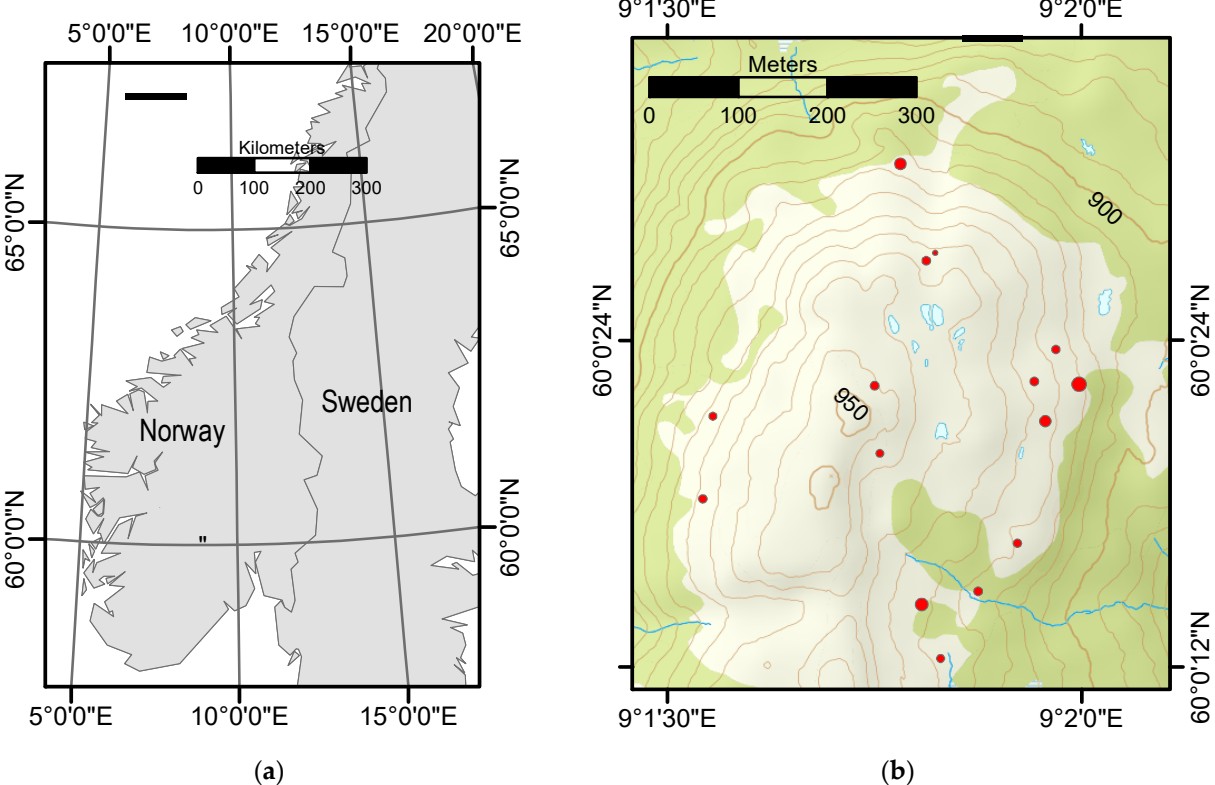

(**a**)　　　　　　　　　　　　　　　　　　　　　　　　　(**b**)

**Figure 1.** (**a**) Location of the study site in southern Norway (black square). (**b**) Selected Norway spruce tree objects (red circles). The different sizes of the red circles represent the downward-looking pyranometer's footprint projected at ground for the different tree objects according to the UAV flight height (see Equation (3)). The green areas are classified as forest areas according to the official N50 topographic map series. Accordingly, the light area is above the treeline. Contour interval is 5 m.

We strived to establish Norway spruce tree objects without disturbances from other tree species. However, due to the generally heterogeneous species composition in the tree-line, some influence of other tree species in the periphery of the footprint of the downward-looking pyranometer used to measure the reflected radiation was unavoidable. This influence was nevertheless considered negligible due to the pyranometer's cosine response. The coordinates of the center of the selected tree objects were registered using a real-time kinematic, differential global navigation satellite system receiver with expected position error (root mean square error) < 0.02 cm [35]. During field work on the 18th of March 2021, crown widths, tree heights and snow depths were measured. For each tree object, snow depth was calculated as the average of three snow depth measurements conducted at 2 m distance from the tree stem in the northern, eastern and western cardinal directions (Figure 2). According to visual observations, this distance was considered suitable to assure that snow depth measurements were performed outside the zone influenced by the tree crown. Thus, the snow depth measurements were likely unaffected by potential snow melting or snow accumulation due to the branches. Tree heights above the snow surface were measured by using a height pole and a vertex clinometer for small and tall tree objects, respectively. The total tree height for each tree object was calculated as the sum of the tree height measurement and the mean snow depth (Figure 2). Crown widths were measured in the east–west direction, since this directional cross-section was considered most important for the trees' snow masking according to the solar azimuth angle. The measurements were conducted at the top of the snow surface.

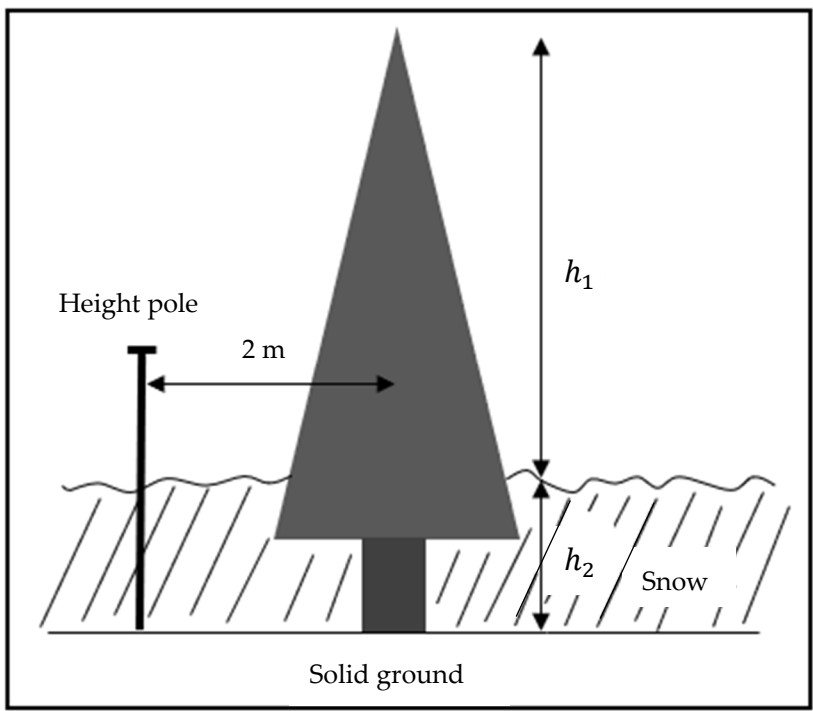

**Figure 2.** Schematic showing tree height and snow depth measurements. Height $h_1$ represents tree height measurements above the snow surface measured by a height pole or a vertex clinometer. Height $h_2$ represents the snow depths measured by a height pole at 2 m distance from the tree stem in the northern, eastern and western cardinal directions. Total tree height was the sum of $h_1$ and $h_2$.

As noted above, tree structural variables played a prominent role in this study because the properties of the trees were expected to control the single-tree albedo. We defined two classes of variables designated "tree structural variables", namely (1) tree height and crown width measured on the ground (see above) and (2) metrics calculated from airborne laser data (see details in Section 2.4). Descriptive statistics for some of the tree structural variables are given in Table 1.

**Table 1.** Mean, range and standard deviation of tree height, crown width in east–west direction and first echo cover index (FCI) of Norway spruce tree objects. See Section 2.4 for detailed description of FCI.

| Tree Structural Variable | Mean | Range | Standard Deviation |
|---|---|---|---|
| Tree height (m) | 4.31 | 2.42–6.70 | 1.07 |
| Crown width (m) | 2.12 | 1.08–3.38 | 0.63 |
| FCI | 0.56 | 0.17–0.76 | 0.18 |

## 2.2. UAV Platform for Measuring Albedo

The UAV platform consisted of a downward-looking SP-610-SS and an upward-looking SP-510-SS pyranometer (Apogee, Logan, UT, USA) mounted on a Matrice 210 V2 quadcopter (DJI, Shenzhen, China) with a SafeAir M200 parachute safety system (ParaZero, Beer Sheva, Israel) on top (Figure 3a). Both the upward-looking (field of view of 180°) and the downward-looking pyranometer (field of view of 150°) were second-class (also known as class C) sensors, with a detector response time of 0.5 s and sensitivity of $0.057\,\text{mVW}^{-1}\,\text{m}^{-2}$ and $0.15\,\text{mVW}^{-1}\,\text{m}^{-2}$, respectively. The reflected radiation was measured by the downward-looking pyranometer (spectral range of 295–2685 nm), which was mounted to a PIXY U three-axis gimbal (Gremsy, Ho Chi Minh City, Viet Nam), and thus made easily attachable to the UAV. The gimbal ensured an absolute horizontal levelling of the downward-looking pyranometer, which was crucial for exact control of the surface of

interest for which the reflected radiation was measured. In addition, the gimbal ensured an unobstructed field of view for the downward-looking pyranometer without disturbances from the drone legs (see Figure 3a). Even though some of the tree objects were growing in sloping terrain, horizontally measured reflected radiation has been shown to be the best representation when providing single-tree albedo at fine spatial resolution [32]. A special carbon mounting bracket attached to the left drone leg attachment was designed for the upward-looking pyranometer (spectral range of 385–2105 nm). This ensured an unobstructed field of view without conflicting with the parachute safety system on top of the UAV. Both pyranometers were connected to AT-100 microCache Bluetooth loggers (Apogee, Logan, USA). This logger was favorable due to its low weight of 0.052 kg and the easily changeable sampling frequency. Since the parachute safety system disturbed the signal strength of the internal GPS receiver of the UAV, an external DJI GPS kit (for Matrice 200 series V2) was attached to the right drone leg. This ensured strong GPS signals, which was necessary when performing preprogrammed mission flights. The positioning mode of the UAV, with strong GPS signals and forward and downward vision systems enabled, had a vertical and horizontal hoovering accuracy according to the producer specifications of $\pm 0.1$ m and $\pm 0.3$ m, respectively.

According to the effective half field of view ($\omega$) of the downward-looking pyranometer, the radius ($r$) of the footprint for which the reflected radiation was measured is given by:

$$r = h_{UAV} \times \tan(\omega) \tag{1}$$

Here, $h_{UAV}$ represents the height of the UAV above the snow surface. Due to the cosine response of the downward-looking pyranometer, we introduced a percentage contribution factor ($f$) to make it easier to control the surface for which most of the reflected radiation of interest was measured. Here, $f$ represents the size of the area of interest relative to the size of the total footprint of the downward-looking pyranometer for which the percentage contribution of the measured reflected radiation is reflected from:

$$f = \frac{\sin(\omega)}{\sin(75°)} \times 100 \tag{2}$$

Here, 75° represents half of the field of view of the downward-looking pyranometer and $\omega$ is the effective half field of view of interest. Solving Equation (2) with respect to $\omega$ and substituting the expression into Equation (1) gives the following radius ($r$) of the footprint of each tree object projected at the snow surface:

$$r = h_{UAV} \times \tan[\sin^{-1}(\frac{f}{100} \times \sin(75°))] \tag{3}$$

where $h_{UAV}$ represents the UAV flight height above the snow surface. In this study, the flight height above canopy was set to 0.7 m and the contribution of interest was set to 80%. Examples of radii of 80% contribution footprints for different tree heights and snow depths are shown in Table 2. It should be noted that the snow depths used in all calculations related to the UAV-based measurements were measured concurrently with the UAV campaign (see details in Section 2.3), since the snow depths were subject to substantial changes during late winter and early spring due to wind and some snow melting in southern sloping locations.

**Table 2.** Footprint area (m$^2$) for 80% footprints of the downward-looking pyranometer for typical tree heights (m) and snow depths (m) included in the study.

| Snow Depth | Tree Height | | |
|---|---|---|---|
| | 2.0 (m) | 5.0 (m) | 8.0 (m) |
| 0.5 (m) | 22.9 | 124.7 | 314.2 |
| 1.0 (m) | 13.9 | 102.0 | 277.6 |
| 1.5 (m) | 6.2 | 81.7 | 243.3 |

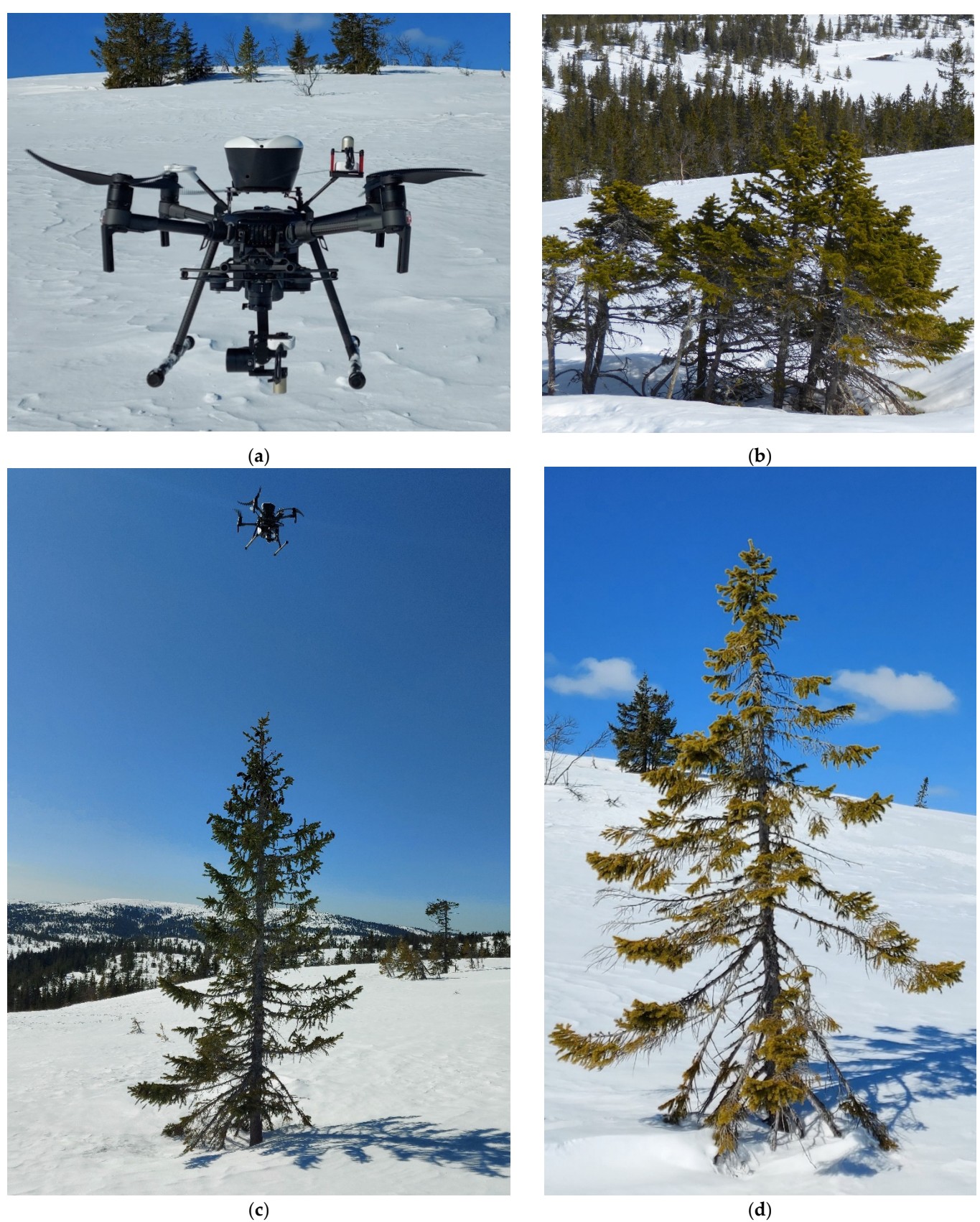

**Figure 3.** (**a**) Set-up of the UAV platform showing the upward- and downward-looking pyranometers, parachute safety system and external GPS receiver. (**c**) UAV in hoovering mode above one of the tree objects. (**b**,**d**) Two extremes of tree crown shape and crown width for two of the tree objects.

The snow-masking effect of each tree object was estimated by assuming the trees had a perfect conical shape (Figure 2), i.e., projecting a triangular shadow area ($A_{shadow}$) on the snow surface. Here, $A_{shadow}$ was calculated as:

$$A_{shadow} = \frac{\tan(\theta)}{2} \times d_{crown} \times h_{projection} \times \left[ 1 + \frac{\sin(\sigma) \times \sin(\theta) \times \cos(\delta)}{\sin(90° - \theta - \sigma)} \right] \quad (4)$$

Here, $\theta$ is the solar zenith angle, which was calculated according to [36] and is given in the Appendix; $\sigma$ and $\delta$ represent the slope angle of the terrain and the difference in direction between the terrain aspect and solar azimuth angle, respectively. Northerly slopes are assigned positive values of $\sigma$, while southerly slopes are assigned negative values of $\sigma$ when assuming the sun is due south. All angles are given in degrees. Terrain slope and aspect angles around each tree object were calculated from the official national detailed terrain model (see Section 2.4), while the solar azimuth angle was derived according to [36], given in the Appendix A. Here, $d_{crown}$ represents the crown diameter in the east–west direction at the top of the snow surface and was estimated for each tree object based on measured crown diameter and snow depth measured concurrently with the UAV campaign (see above); $h_{projection}$ represents the projection height of the tree objects, which was calculated by subtracting the field-measured tree height by the current snow depth. The last factor in Equation (4) represents the relative change in snow-masking area for sloping terrain [37], thus it equals unity for trees growing on a horizontal surface ($\sigma = 0$).

### 2.3. Radiation Data

Radiation measurements were conducted from 15th of April to 19th of April 2021, during days with sunny and clear sky conditions and without any snow interception on the trees. The flights were performed from morning at 09:45 (UTC +2) as the earliest to evening at 18:30 (UTC +2) as the latest time of day. Flights were performed using a DJI Controller Cendence connected to a DJI CrystalSky monitor. Mission flights were preprogrammed using the DJI Pilot app. This software facilitates completely repeatable flights, for which the autonomous mission flights can be controlled manually at any time during the flight mission. The 15 tree objects were arranged into four preprogrammed mission flights, where neighboring tree objects were measured in the same flight mission. This arrangement was necessary due to limited power supply of the UAV and to ensure that visual contact was maintained between the UAV and the observer for each flight mission. The UAV was programmed to hover for one minute above each tree object (Figure 3c), for which the sampling frequency was set to 1 Hz. During the automatic mission flights, it was observed visually several times that heading for positioning above the tree objects was too inaccurate. This caused the UAV to be incorrectly positioned above the tree objects. When it was discovered visually that the UAV's hoovering position deviated about 2–3 m from the positions of the tree objects, the hoovering position was manually corrected so the UAV was centered directly above the tree objects. Flight data records were retrieved by uploading the flight logs to the online DJI Flight Log Viewer at the Phantom Help website (https://www.phantomhelp.com/LogViewer/Upload/ (accessed on 1 January 2022)). Radiation data stored on the Apogee Bluetooth loggers were downloaded via the Apogee Connect app. Collected radiation data for each tree object were retrieved by conducting a manual matching process of flight records and radiation data based on the latitudinal, longitudinal and vertical hoovering positions of the UAV and approximately noted hoovering times for the different tree objects. Visual inspection of the reflected radiation for each tree object revealed unstable observations for typically the first 10 s of the hoovering time. Therefore, these were omitted from the one-minute measurement series to allow stabilization of the pyranometers at each tree object. The albedo of each tree object was calculated as the median of the reflected radiation divided by the incident radiation.

### 2.4. Airborne Laser Scanner Data

Airborne laser scanner (ALS) data were acquired on 18 June 2017, under leaf-on conditions using a fixed-wing aircraft. The dataset was used by the data vendor (Terratec AS, Norway) to produce the official national detailed terrain model by classifying the points as ground and non-ground echoes using the progressive triangular irregular network (TIN) densification algorithm [38] implemented in TerraScan software [39]. Terrain aspect and slope around each tree object were calculated by area-weighting the aspect and slope angles within a radius corresponding to 80% of the footprint of the downward-looking pyranometer when projected onto the ground surface. However, to mimic the smoothing effect of the snow resulting from the winds' re-distribution of the snow during the winter season, it was found to be appropriate to smooth the TIN terrain model to obtain approximately similar sizes of the 80% footprint projection and the triangular facets of the TIN used to characterize terrain slope and aspect. After an iterative trial and error process with different smoothing levels using linear interpolation, this condition was met when the vertices of the initial TIN model were allowed to deviate up to 2 m from the smoothed TIN model. This resulted in a thinning for which the 80% footprint projection consisted on average of 1.8 and 1.6 triangular facets for aspect and slope, respectively. The average area-weighted standard deviation of slope and aspect angles for all 80% footprint projections were 0.9° and 5.3°, respectively. The average slope angle was 10.6°. The maximum slope angle was 15.9°.

As noted in Section 2.1, one of the classes of tree structural variables used in the analysis was composed of metrics calculated from the ALS data. The raw ALS data covering our study area were acquired by parallel flight lines with side overlap between adjacent strips, plus a single perpendicular flight line crossing the main flight direction of the ALS block of strips. Prior to calculating these tree structural variables, harmonization of the point density of the ALS data was considered important. Order statistics, such as the maximum height, are monotone increasing functions of numbers of points for a given target area [40]. The maximum height is a relevant candidate as a tree structural variable (see below). To keep the point density stable across the study area, all data from the perpendicular flight line and from the overlap zone between adjacent parallel strips were discarded. The resulting minimum point density for our study area was 5 points m$^{-2}$. Normalized height values were computed for all echoes relative to the official TIN by linear interpolation. All classified ground and non-ground points with negative normalized height values were assigned the value zero. All classified ground points were assumed to lie on the official terrain surface and were assigned the value zero.

For each tree object, common tree structural variables were calculated based on all first and single return echoes within a radius of 2.4 m. A threshold height of 0.2 m was adopted in the calculation. This threshold height was found to be reasonable due to the small sizes of the studied tree objects and testing with different threshold heights in the range of 0.2–1.3 m prior to the final analysis. Before selecting a radius of 2.4 m, different radii were tested. The objective was to select the radius of a circle corresponding to the maximum crown width to ensure that the calculated structural variables captured properties of the entire tree crown. Because the crowns were not perfectly circular, a theoretical maximum radius of 1.7 m (Table 1) resulted in insufficient coverage of the largest crown. Among the tested radii, 2.4 m was found to be appropriate as a fixed radius for all tree objects. It should be noted that we also tried a strategy with the dynamic radius depending on the height of the tree objects reflecting the footprint of the downward-looking pyranometer. However, this strategy did not improve the results beyond the more simplistic approach.

The tree structural variables included the maximum height ($h_{max}$), average height ($h_{avg}$), standardized standard deviation of the height ($h_{std}$), 50% height percentile ($h_{p50}$), 90% height percentile ($h_{p90}$) [41] and the first echo cover index (FCI) [42]. Here, $h_{std}$ is a height dispersion variable, for which a large value corresponds to the distribution of the canopy biological matter across a large part of the tree's vertical extent. FCI is the fraction of first or only canopy echoes above a certain height threshold to all first echoes and serves



as a proxy for canopy density. FCI is also closely related with the metric commonly used to model LAI with ALS data [43]. Thus, it reflects the density of the biological matter.

### 2.5. Statistical Analysis

Repeated radiation measurements were performed 6 to 7 times during different times of day. To account for the correlations between observations within each tree object, linear mixed effects models (LMMs) were used to estimate the dependency between the albedo and the tree structural variables. The LMMs were fitted using the lme4 package [44] in R software (Version 4.1.2, Vienna, Austria) [45] by using the restricted maximum likelihood (REML) [46] method. Since the main focus was to calculate variance components, and REML produces (usually) unbiased variance components as opposed to the biased maximum likelihood variance estimates (e.g., [47]), the REML algorithm was considered more appropriate for modeling of the relationship between the albedo and the tree structural variables. LMMs were fitted for each structural variable one at a time, with tree object locations included as random intercept effects. In addition, to account for the diurnal effect of the sun, the interaction effects between the tree structural variable and the solar zenith angle were included as fixed effect. The solar zenith angle was assigned negative values before solar noon. The dependency between the tree structural variables and the albedo was reported in terms of the marginal $R^2$ ($R_m^2$) [48]. $R_m^2$ can be interpreted as the proportion of the albedo variability explained by the fixed effects in the model. In addition, the mean absolute error (MAE) and the proportion of variance of the random effect were calculated for each of the fitted LMMs. The latter of the two assesses the differences between tree objects in explaining the variation after accounting for the fixed effects' variance.

## 3. Results

### 3.1. Solar Angle and Snow-Masking Effects on Albedo

A large variation of solar zenith angles was present, ranging from –65° to –50° before solar noon (indicated by negative sign) and 53° to 73° after solar noon. Overall, the albedo varied from 0.39 to 0.99. The within-tree object variation of the albedo was more prominent than the between-tree object variation (Figure 4), showing the importance of the diurnal effect of the sun. Person correlation coefficients between the albedo and the solar zenith angle and solar azimuth angle were –0.45 and –0.48 (Table 3), respectively. For the terrain slope and aspect, the Pearson correlation coefficients were –0.36 and 0.17 (Table 3), respectively. To account for the temporal effect caused by the diurnal course of the sun, LMMs for the terrain slope and aspect were fitted one at a time with both solar azimuth and zenith angles and their interactions with the terrain. These models explained 44% and 49% of the variation in albedo for the slope and aspect, respectively. Spruce trees growing on easterly slopes had the largest albedo in the early morning for large negative solar zenith angles and easterly solar azimuth angles (Figure 4). In the evening, when having large solar zenith angles and westerly solar azimuth angles, the albedo values for these trees were the smallest. Opposite effects were revealed for spruce trees growing on westerly slopes. The snow-masking effect of the trees, as estimated by Equation (4), was found to be statistically significant for the albedo, with a Pearson correlation of –0.45 (Table 3). The snow-masking effect was the on-ground variable that explained most of the albedo variation. When just projecting the tree object shadow onto a horizontal plane without accounting for the diurnal interaction between the sun and the sloping terrain, the Pearson correlation between the albedo and the snow-masking was reduced to a value of –0.25. For spruce trees growing on northerly or southerly slopes (within ±45° from the north-south cardinal direction), the diurnal trend caused by the interaction between the terrain aspect and the diurnal course of the sun was weaker. For spruce trees growing on northerly and southerly slopes, the mean diurnal albedo range was 0.23, while for spruce trees growing on easterly and westerly slopes, the mean diurnal albedo range was 0.41. However, it is important to emphasize that only 1/3 of the trees included in this study were growing on northerly or southerly slopes.

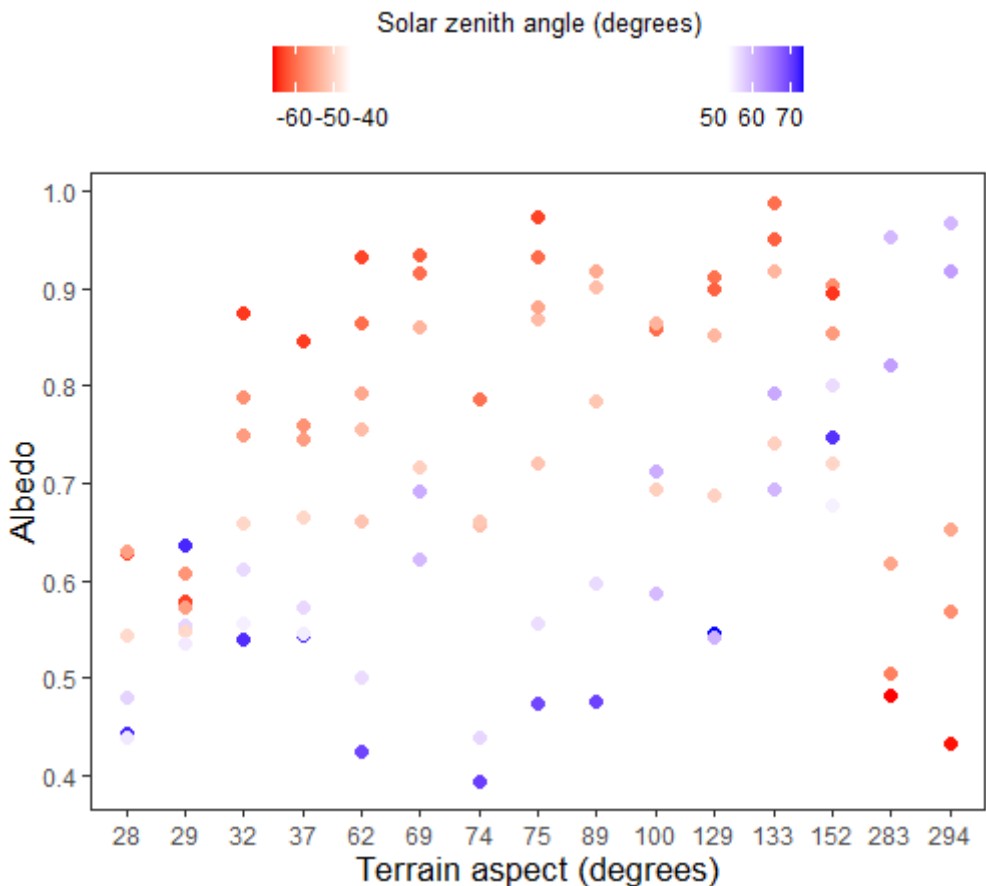

**Figure 4.** Albedo observations for each of the 15 Norway spruce tree objects, represented by their terrain aspect angles (defined to true north). The color grading represents the solar zenith angle (in degrees), where negative angles correspond to measurements before solar noon.

**Table 3.** Pearson correlation coefficients and corresponding *p*-values between albedo and solar angles, terrain slope angles and the snow-masking of the tree objects ($A_{shadow}$).

| Explanatory Variable | Pearson Correlation Coefficient | *p*-Value |
|---|---|---|
| Solar zenith angle | −0.45 | $4.10 \times 10^{-6}$ |
| Solar azimuth angle | −0.48 | $1.17 \times 10^{-6}$ |
| Terrain slope angle | −0.36 | 0.0004 |
| Terrain aspect angle | 0.17 | 0.0924 |
| $A_{shadow}$ | −0.45 | $5.19 \times 10^{-6}$ |

### 3.2. Tree Structural Effects on Albedo

The Pearson correlation coefficients (Figure 5) revealed that the albedo was negatively correlated with all the tree structural variables, showing that both taller and denser tree objects and tree objects with increasing height dispersion decreased the albedo. However, when not accounting for the diurnal effect of the sun, FCI was the only statistically significant tree structural variable (Pearson correlation coefficient with *p*-value < 0.001), except for the snow-masking effect. All height variables calculated from the ALS data explained more of the variation in albedo than the field measured tree height. When accounting for the diurnal effect of the sun and the correlation for repeated albedo measurements within the same tree object, the tree structural variables explained 19–28% of the variations in albedo (Table 4). All LMMs had a mean absolute error of ~0.1. This corresponded to residuals of a magnitude of 10–25% of the measured albedo. For the LMMs, FCI (*p*-value = 0.0334) was the only structural variable found to be statistically significant.

However, all interaction effects between each of the tree structural variables and the solar zenith angle were statistically significant ($p$-values < 0.05). For all tree structural variables except FCI, the differences between tree objects explained 18.7–23.2% of the variance in albedo (Table 4) after accounting for the variance explained by the tree structural variables themselves and the interactions between the solar zenith angle and the tree structural variables. For the FCI, the differences between tree objects explained only 6.7% of the remaining variance in albedo after accounting for the variance explained by the fixed effects (Table 4). In summary, these results demonstrate that the characteristics of the tree objects themselves did not explain much of the variation in albedo in this study.

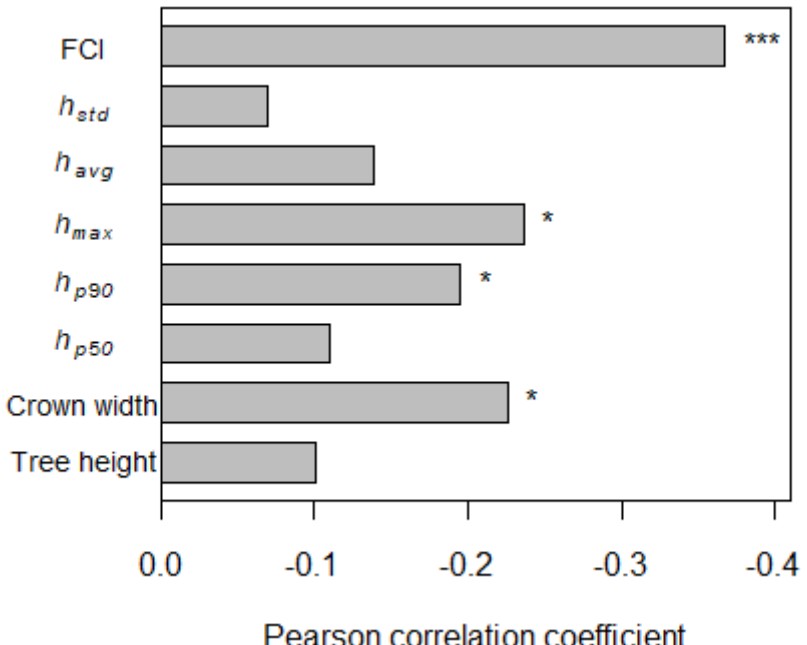

**Figure 5.** Pearson correlation coefficient between albedo and tree structural variables. Note: * indicates $p$-values < 0.1, while *** indicates $p$-values < 0.001.

**Table 4.** Marginal $R^2$ ($R^2_m$), mean absolute error (MAE) and proportion of variance (in %) explained by differences between tree objects after accounting for the fixed effects variance for the LMMs fitted for each of the tree structural variables, one at a time.

| Tree Structural Variable | $R^2_m$ | MAE | % Variance Explained by Difference between Trees [1] |
|---|---|---|---|
| Tree height | 0.23 | 0.0916 | 23.2 |
| Crown width | 0.27 | 0.0929 | 19.4 |
| $h_{p50}$ | 0.23 | 0.0932 | 23.2 |
| $h_{p90}$ | 0.28 | 0.0911 | 21.3 |
| $h_{max}$ | 0.27 | 0.0940 | 18.7 |
| $h_{avg}$ | 0.24 | 0.0926 | 22.7 |
| $h_{std}$ | 0.19 | 0.0956 | 22.5 |
| FCI | 0.28 | 0.1042 | 6.7 |

[1] After accounting for the variance explained by the fixed effects.

## 4. Discussion

This study presented, for the first time, fine-spatial, single-tree albedo measurements of Norway spruce collected by a UAV platform in a heterogeneous boreal–alpine treeline. Earlier studies have reported tower-based wintertime albedo observations for spruce forests in the range of 0.11 to 0.33 [16,22,49,50]. Webster and Jonas [22] reported albedo values measured from a UAV over Norway spruce trees with tree heights of 11 to 35 m in the

range of 0.06 to 0.52. They found that smaller canopy densities and discontinuous forests dominated by open gaps and large snow interceptions on the canopy gave the largest albedos. The range of albedo observations reported in our study did not match those in previous studies, but this was not expected, since we performed single-tree measurements in open terrain. Thus, we are unaware of any previous results that might serve as basis for comparison to our observations. Typical albedo values for even, non-polluted snow surfaces are 0.96–0.98 (e.g., [19]), while 16% of our observations had albedos larger than 0.9. Possible reasons for the large albedo values reported in the current study will be discussed in detail later in this section.

### 4.1. Statistical Assessment of LMMs

The statistical analysis showed that all tree structural variables were negatively correlated with the albedo; however, only the FCI was found to have a statistically significant effect (both the Pearson correlation and as a fixed effect in the LMM). The error distribution of the LMMs revealed a slight tendency of deviations from normality (not shown here). However, this was not considered critical for modeling the LMMs' correlation coefficients. In addition, it has been shown that significance tests assuming Gaussian errors in which the LMMs' random effect accounted for non-independence in the data remain fairly reliable for non-normality [51]. The relatively large MAEs computed for the LMMs may be due to the small explanatory power of the tree structural variables. Additionally, the small difference in marginal $R^2$ for the different LMMs indicated minor contributions to albedo variation among the different tree structural variables. This was also supported by the generally small portion of variance of the albedo explained by the differences between tree objects after accounting for the fixed effects.

### 4.2. Assessment of Tree Structural Variables in Relation to Albedo

All ALS-derived height variables showed stronger correlations with the albedo than the field-measured tree height. However, comparing the effects of the different height variables should be done with caution, since the Pearson correlation was statistically significantly different from zero at the 90% level for only one of the variables ($h_{max}$). It is also important to emphasize that for such small trees as typically found in the treeline, the measurement errors in the ALS data introduce random variation of a potentially substantial magnitude relative to the size of the objects being measured. With larger trees, greater ranges of tree heights and larger pulse densities of the ALS data, we would likely have detected stronger relationships between the albedo and the ALS tree structural variables by increasing the relative precision of the variables used to explain the albedo variation. As opposed to the summertime boreal albedo [23], the ALS height variable $h_{std}$ was found to be unrelated with the albedo. It is, therefore, reasonable to assume that the size of the tree objects in this study was too small to effectively influence the albedo through the mechanism of mutual scattering and reflections by their three-dimensional canopy structure.

The absence of correlation between albedo and tree height measured in the field could be explained by the distinct crown shapes of the spruce trees. Typically, spruce trees have conical shapes with narrow, thin treetops (Figure 3d). The upper parts of these trees, therefore, have limited amounts of biological matter that could interact with the incoming radiation, which could affect the albedo. Only when the trees grow in clusters and form thickets (Figure 3b) can the treetops and upper parts of the tree canopy effectively influence the albedo. It is, therefore, not surprising that especially field-measured tree height of solitary small trees is a poor covariate for albedo. For laser data on the other hand, it is well known that laser beams tend to penetrate into the tree crowns before a sufficient amount of emitted light is reflected to trigger an echo (e.g., [52]). The maximum echo height value will typically be smaller than the actual tree height. This was illustrated by the calculated mean difference between the measured tree height and $h_{max}$ of 0.96 m. The distribution of the ALS echoes from which the ALS variables were calculated reflects the distribution of the biological matter. In the same way as for the active laser beams, the distribution

of the biological matter also controls the reflections of passive solar radiation. It was, therefore, reasonable that the ALS height variables correlate better with the albedo than the field-measured tree height, as found in our study. This finding is interesting, showing how intricate the relationships are between the tree size and the distribution of biological matter within the tree crowns and the spatial resolution of the radiation measurement. Further, since data from laser scanners can potentially be useful as covariates for explaining albedo in a small-tree environment, structural variables calculated from ALS can perhaps even be suitable as explanatory variables in prediction modeling.

### 4.3. Challenges of Single-Tree Albedo Measurements by UAV

Our results demonstrated that single-tree boreal–alpine Norway spruce albedo showed weaker linear relationships with tree structural variables than previously reported (e.g., [22,23]). This was somewhat unexpected, because ground covered by snow gives a larger spectral contrast than typical ground vegetation in boreal forests and boreal–alpine treelines [25]. Five reasonable causes are likely to explain the weak relationships between the albedo and the tree structural variables in this study: (1) strong solar diurnal effect strengthened by (2) increased reflection from sloping terrain outside the footprint of the downward-looking pyranometer, (3) radiation measurement errors for large solar zenith angles and (4) failure to restrict the spatial resolution of the reflected radiation to match the tree object size of interest, aggravated by (5) imprecise heading for the hoovering position of the UAV above the tree objects. Below, we elaborate these five causes in detail.

### 4.3.1. Effects of Diurnal Solar Course and Sloping Terrain

Regarding the solar diurnal effect (1), strong diurnal and systematic differences in albedo were detected for tree objects growing on slopes with different terrain aspect angles. In the morning, with easterly solar azimuth angles and large negative solar zenith angles, spruce trees on easterly slopes gave their largest albedo values, while spruce trees on westerly slopes gave their smallest albedo values. This was controlled by the interactions between the sun's position and the terrain slope and aspect, for which the normal of the east-facing slopes was more perpendicular to the incoming radiation in the morning, while the angle between the sun and the normal of the west-facing slopes was relatively large. Opposite effects were found in the afternoon and late evening. These diurnal trends were consistent with previous single-tree albedo observations in sloping terrain [32] and for albedo observations in sloping terrain without trees [32,53,54]. For spruce trees growing on northerly or southerly slopes, the diurnal trend and within-plot variation were less prominent. This was reasonable, because the solar zenith angle was smallest (in absolute value) when the directional difference between the terrain aspect and solar azimuth angle was largest or smallest, thereby weakening the shadow effect caused by the terrain slope. Except for FCI, only the LMMs interaction effects between the solar zenith angle and the tree structural variables were statistically significant at the 95% level. This substantiates the control of the sun's position on the albedo variation. Webster and Jonas [22] reported similar findings for moderate solar zenith angles (ranging from 35 to 55°), whereby correlations between albedo and canopy heights and ground view fractions were present, but the correlation weakened and disappeared for large solar zenith angles (66°).

It is likely that the diurnal effect on the albedo in our study was amplified by (2) reflections from sloping neighboring terrain outside the footprint of the downward-looking pyranometer. Especially when the sun's azimuth moves into a position aligned with the terrain aspect, the predominant forward direction of reflectance [55] amplifies the reflected radiation. In addition, the increased tendency of anisotropic reflection for large solar zenith angles (in absolute value) increases the albedo. These effects reasonably explained why one tree object growing on an easterly slope (slope angle of 9° and aspect angle of 89°) was detected with two albedo observations (not included in the analysis) of 1.12 and 1.03 for times at 10:13 and 10:39, respectively. Such observations are a typical consequence of sloping terrain [19].

### 4.3.2. Effects of Measurement Errors for Large Solar Zenith Angles

Radiation measurement errors (3) should be expected for measurements collected during large solar zenith angles. The directional response of the upward-looking pyranometer was less than 30 Wm$^{-2}$ for solar zenith angles up to 80°. For the downward-looking pyranometer, the directional response was less than 20% for solar zenith angles of up to 60°. One-third of the albedo observations from our study had solar zenith angles >60° (in absolute value). Due to the high northern latitude of the study site, large solar zenith angles were unavoidable when providing albedo observations for wintertime conditions. This likely caused errors in the measured radiation, mainly due to the pyranometers' deviation from the cosine law, making them less sensitive at large solar zenith angles [56]. The albedo values reported in this study were likely to suffer from overestimation of the reflected radiation due to increased reflections inside the pyranometer dome [57]. Additionally, there was likely an underestimation of measured incoming radiation, because of the reflection properties of the sensor's black paint [58,59], and especially during conditions dominated by direct radiation [58]. Both mechanisms contribute to an overestimation of the albedo values, and partially explain the large albedo observations reported here.

### 4.3.3. Effects of Imprecise Spatial Resolution and Heading of the Hoovering Position

Earlier research has demonstrated that albedo measurements for single trees with dense canopies should be provided with horizontal orientation of the pyranometers in sloping terrain [32]. However, the choice of either horizontal or slope-parallel orientation of the sensors also depends on the spatial size of the footprint of the downward-looking pyranometer relative to the tree size of interest. In the present study, the footprint area (Table 2) for the tallest tree objects with narrow treetops was as much as 10 times the size of the horizontal projection of the tree crown. Since the interaction between the diurnal course of the sun and the sloping terrain was the main factor driving the albedo, it is reasonable to believe that (4) the spatial size of the footprints for which the reflected radiation was measured should have been smaller. In doing so, the diurnal course of the albedo would have revealed a more typical minimum around solar noon, being less dependent on the aspect of the sloping terrain. The large albedo observations reported here were likely a result of an excessively large contribution of the snow surface around the tree objects, which dominated the response at the expense of the reflected radiation of the tree objects themselves. However, we did not detect smaller within-tree object variation in albedo observations or stronger relationships between albedo observations and tree structural variables for smaller tree objects having considerably smaller footprints for the reflected radiation. Therefore, it is reasonable that the failure to restrict the spatial resolution of the reflected radiation was compounded by (5) imprecise heading for the hoovering position of the UAV above the tree objects. According to the producer specifications, the UAV had a horizontal hoovering accuracy of ±0.3 m. However, it was observed several times that the heading for the positioning above the tree objects was too inaccurate during the automatic flights. This caused the UAV to be incorrectly positioned above the tree objects. When it was discovered visually that the UAV's hoovering position deviated about 2–3 m from the position of the tree objects, the hoovering position was manually corrected so the UAV was centered directly above the tree objects. Accordingly, some errors should be expected in the albedo measurements due to non-consistent hoovering positioning for the repeated flights. The lack of accurate positioning of the UAV weakened the relationship between the tree structural variables and the measured albedo by increasing the influence of the surrounding snow surface.

### 4.4. Future Perspectives

The use of a UAV with a high-precision navigation system to improve the heading hoovering position will be useful for albedo measurements at fine spatial resolutions of small single trees. To limit the extent of the footprint of the reflected radiation, a new method has been proposed for field-measured albedo by using a black cylindrical shade

curtain to block the radiation from the surroundings [60]. However, when providing fine-spatial, proximal remotely sensed albedo, as with our UAV, this method is not applicable. For this purpose, a field-stop device that limits the field of view of the downward-looking pyranometer would be useful, such as field-stop spectral reflectance sensors used to provide individual tree measurements of photochemical canopy reflectance [61]. This would also minimize the measurement errors appearing at large solar zenith angles [58]. Assuming that such modifications are feasible, UAVs providing fine-spatial radiation measurements could become important tools for collecting stand-level- and tree-level-specific albedo.

Datasets and effective methods of fine-spatial vegetation sampling describing tree and forest characteristics already exists, e.g., such as ALS, satellite-based remote sensing and land-based high-resolution vegetation mapping. However, albedo data and effective strategies for providing fine-spatial radiation measurements matching this spatial resolution are still missing. Fine-spatial albedo measured by UAV will contribute important information on how structural metrics for different species and different vegetation communities are related to albedo, and will help close this knowledge gap. Particularly, fine-spatial UAV-measured albedo will improve the representation and parametrization of the snow-masking of trees in open terrain and discontinuous forests in larger-scale models [22]. This will be useful for the evaluation of the climatic effects of treeline migration. Additionally, detailed knowledge about the relationship between tree-specific variables and albedo will be important for forest–climate mitigation strategies in boreal forests.

## 5. Conclusions

The use of UAV platforms equipped with pyranometers represents a new method for proximal remotely sensed albedo measurements. This enables an opportunity for the flexible collection of fine-spatial albedo data in remote and heterogeneous environments over larger spatial extents. In this study, we provided single-tree wintertime Norway spruce albedo in a boreal–alpine treeline in Norway. The trees' snow-masking effect was statistical significantly (at 95% level) related to the albedo and was the single on-ground variable explaining most of the albedo variation. Among the tree structural variables, the FCI had strongest relationship with the albedo and was the only tree structural variable that was statistically significant (at 95% level). Even though the wintertime albedo measurements were performed in mid-April, the high latitude of the field site was challenging due to the low sun position. Thus, the diurnal effect of the sun in interaction with the sloping ground snow surfaces was the most important factor driving the albedo. For the further development of UAV-measured fine-spatial albedo, we recommend the use of UAVs connected to high-precision navigation systems and a field-stop device on the downward-looking pyranometer to limit the field of view of the reflected radiation.

**Author Contributions:** Conceptualization, E.N.R.; methodology, E.N.R., T.G. and E.N.; data collection, E.N.R.; software, E.N.R. and T.G.; validation, E.N.R. and E.N.; formal analysis, E.N.R.; investigation, E.N.R. and E.N.; resources, E.N.R.; data curation, E.N.R. and T.G.; writing—original draft preparation, E.N.R.; writing—review and editing, E.N.R., E.N. and T.G.; visualization, E.N.R.; project administration, T.G.; funding acquisition, E.N. and T.G. All authors have read and agreed to the published version of the manuscript.

**Funding:** This research was supported by the Research Council of Norway (Project #281066: "Changing Forest Areas and Forest Productivity—Climatic and Human Causes, Effects, Monitoring Options and Climate Mitigation Potential", Project #302701: "Climate Smart Forestry Norway") and the Norwegian University of Life Sciences.

**Data Availability Statement:** The field data are not publicly available due to the privacy of the private landowners. The UAV data are available on request from the corresponding author. The ALS data (3rd party data) are available at https://hoydedata.no/ (accessed on 1 January 2022).

**Conflicts of Interest:** The authors declare no conflict of interest. The funders had no role in the design of the study; in the collection, analyses, or interpretation of data; in the writing of the manuscript, or in the decision to publish the results.

## Appendix A

The cosine to the solar zenith angle ($\theta$) in radians is given by:

$$\cos(\theta) = \sin(L)\sin(\varphi) + \cos(L)\cos(\varphi)\cos(h), \tag{A1}$$

where $L$ represents the latitude in degrees, $\varphi$ represents the solar declination angle in radians and $h$ is the hour angle in radians, which is dependent on the latitude, longitude and standard meridian of the local time zone. The declination angle and hour angle are calculated as [36]:

$$\varphi = 0.409\sin\left(\frac{2\pi}{365}N - 1.39\right), \tag{A2}$$

$$h = (12 - t_{solar})\frac{\pi}{12} \tag{A3}$$

Here, $N$ is the day and the solar time ($t_{solar}$) in hours is given by:

$$t_{solar} = LST + \frac{E}{60} + \frac{(SM - LOB)}{15}, \tag{A4}$$

where $LST$ represents the local standard time in hours, $E$ stands for the equation of time, $SM$ is the standard meridian of the local time zone in degrees and $LOB$ is the longitude of the observer in degrees. The equation of time in minutes is given by:

$$E = (5.0323 - 430.847\cos t + 12.5024\cos 2t + 18.25\cos 3t - \\ 100.976\sin t + 595.275\sin 2t + 3.6858\sin 3t - 12.47\sin 4t)/60, \tag{A5}$$

where $t$, in radians, is given by:

$$t = \frac{2\pi}{366}N + 4.8718 \tag{A6}$$

The azimuth angle ($\gamma$) in radians is calculated by [36]:

$$\cos(\gamma) = \frac{\cos(\theta)\sin(L) - \sin(\varphi)}{\cos(L)\sin(\theta)} \tag{A7}$$

where $\theta$, $L$ and $\varphi$ represent the solar zenith angle in radians, the latitude in degrees and the solar declination angle in radians, respectively.

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
