# Peer review of "Fine-Spatial Boreal–Alpine Single-Tree Albedo Measured by UAV: Experiences and Challenges"

_remotesensing, doi:10.3390/rs14061482_

Round 1
Reviewer 1 Report
This manuscript is well organized and well written. I haven't found much to be improved, except Fig. 1.
For Fig. 1, the DEM is suggested to add to the left panel, and the altitude should be written or shown on the right panel rather. Although contour interval is clearly mentioned in the figure text, the readers cannot tell the elevation ranges from the figure.
Author Response
We are grateful to the reviewer for his/her comment.
Please find below the reviewer’s comment and our response to it.
REVIEWER’S COMMENT: This manuscript is well organized and well written. I haven't found much to be improved, except Fig. 1. For Fig. 1, the DEM is suggested to add to the left panel, and the altitude should be written or shown on the right panel rather. Although contour interval is clearly mentioned in the figure text, the readers cannot tell the elevation ranges from the figure.
AUTHORS’ RESPONSE: Thank you. We do not consider it meaningful to add the DEM contours to the left panel since this figure just shows where the study area is located. There is therefor not a purpose for doing so. However, we have added the scale to the left panel. For the right panel, we agree it would be informative to add the altitude to the contours to tell the elevation ranges just from reading the figure.
AUTHORS’ ACTION: We have added the scale to the left panel of Fig. 1 and the altitude to the right panel of Fig. 1.
Reviewer 2 Report
Dear authors,
The manuscript with the title “Fine-Spatial Boreal-Alpine Single-Tree Albedo Measured by 2 UAV: Experiences and Challenges” aimed to demonstrate the experience of the use of UAV for measure single-tree Norway spruce albedo and small trees.
This manuscript is well organized and provides a good explanation of the work for the readers. However, there are a few comments and suggestions about this article that should be addressed:
Abstract:
The abstract is informative and well done.
Introduction:
The authors should say where your work is different. Why this difference should make it better than the other methods? Is it possible to identify a limitation of the state-of-the-art methods, as well?
Finally, in this section authors should explain the results that you expected to obtain.
Materials and Methods:
Line 95: I would suggest switching “situated” to “located”.
Line 103: I would suggest improving this sentence.
Line 120: Please, consider supporting this sentence with one reference.
Line 137: Add manufacturer information (example: DJI, Shenzhen, China). Other example in R (Version 4.3.3, ….). Consider adding this information to the rest of the document.
Line 202: Considerer formatting the table. It can be difficult for the readers to understand.
Line 292: “several times” - How many?
Results:
Lines 313 to 314: Consider adding this sentence on materials and methods (section 2.3- radiation data).
Discussion:
Consider splitting the discussion in small sections in order to help the readers to understand your findings.
In the final of the discussion, the authors should elaborate a little about the future scopes of this work.
Conclusions:
The authors provide a summary of the conclusions and further advice on the use of this type of technology.
Author Response
We are grateful to the reviewer for reading of the manuscript and constructive suggestions for improvements. Please find below the reviewer’s comments and our response to each of the comments.
REVIEWER’S COMMENT: The manuscript with the title “Fine-Spatial Boreal-Alpine Single-Tree Albedo Measured by 2 UAV: Experiences and Challenges” aimed to demonstrate the experience of the use of UAV for measure single-tree Norway spruce albedo and small trees. This manuscript is well organized and provides a good explanation of the work for the readers.
AUTHORS’ RESPONSE: Thank you for kind words, much appreciated.
REVIEWER’S COMMENT: Introduction: The authors should say where your work is different. Why this difference should make it better than the other methods? Is it possible to identify a limitation of the state-of-the-art methods, as well? Finally, in this section authors should explain the results that you expected to obtain.
AUTHORS’ RESPONSE: Thank you. We are unsure if we understand what the reviewer asks for here, but we will try to explain. Both the title and the introduction are quite clear in how our work is different. For the very first time, we provide fine-spatial single-tree albedo measured by UAV. This has never been done before. Our measurement platform is different from other datasets or other measurement methods because of its fine-spatial resolution which can be sampled for larger spatial extents, even in remote areas. This is important for heterogeneous and fragmented vegetation. This is clearly stated in the introduction. Currently, there are no existing methods or sources with regular provisions able to do so, which is the major limitation of the state-of-the-art methods. This is also written in the introduction. Since this study is about a method development and present a preliminary testing, we had no clear opinions or hypothesis of how good or bad our result would be. So, for this study we considered it more appropriate to define the aims of this study, without hypothesizing opinions we did not have. Please, let us know if we misunderstood.
AUTHORS’ ACTION: No action.
REVIEWER’S COMMENT: Line 95: I would suggest switching “situated” to “located”
AUTHORS’ RESPONSE: OK, corrected.
REVIEWER’S COMMENT: Line 103: I would suggest improving this sentence
AUTHORS’ RESPONSE: Good point, we agree.
AUTHORS’ ACTION: We have rewritten the sentence for improving the clarity.
REVIEWER’S COMMENT: Line 120: Please, consider supporting this sentence with one reference.
AUTHORS’ RESPONSE: It is a bit unclear what the reviewer has in mind here. Neither we understand which sentence it is referred to. The sentences before and after just describe what we have done. This is simply based on our decisions of how to perform the field measurements in relation to what the purpose of our research was.
AUTHORS’ ACTION: No action.
REVIEWER’S COMMENT: Line 137: Add manufacturer information (example: DJI, Shenzhen, China). Other example in R (Version 4.3.3, ….). Consider adding this information to the rest of the document.
AUTHORS’ RESPONSE: OK, thanks, corrected.
REVIEWER’S COMMENT: Line 202: Considerer formatting the table. It can be difficult for the readers to understand.
AUTHORS’ RESPONSE: We are unsure what the reviewer finds difficult about this table. Please, be more specific. However, we have decided to move the unit of measurement (m) for each number of snow depth and tree height. Then it should be clear which numbers are given as m and which are in m2. Maybe this helps to understand the table better?
AUTHORS’ ACTION: We have moved the unit of measurement.
REVIEWER’S COMMENT: Line 292: “several times” - How many?
AUTHORS’ RESPONSE: Good point, thank you. Corrected.
REVIEWER’S COMMENT: Lines 313 to 314: Consider adding this sentence on materials and methods (section 2.3- radiation data).
AUTHORS’ RESPONSE: Good idea, we agree. Corrected.
REVIEWER’S COMMENT Discussion: Consider splitting the discussion in small sections in order to help the readers to understand your findings. In the final of the discussion, the authors should elaborate a little about the future scopes of this work.
AUTHORS’ RESPONSE: Good idea. We agree that splitting the text into smaller subsections with their own headings will help on the readability of the discussion. We also think this helps in clarifying our main findings. In the final of the discussion, we have divided a part of the text in its own subsection called “Future Perspectives”. In this section we have also added some points to elaborate future scopes of this work more widely.
AUTHORS’ ACTION: We have divided the discussion into smaller subsections with their own headings. Additionally, we have rewritten the final part to elaborate a little about future scopes of this work.
Reviewer 3 Report
Based on the UAV system, the author puts forward a new method to measure the albedo of Norwegian spruce, which can provide new insights into the fine reflection of the structure of Norwegian spruce. The method and experimental design of the manuscript are reasonable, and the conclusion and discussion are also substantial and reliable. However, there are still some problems that need to be paid attention to. There are a lot of spelling mistakes in the manuscript and most of the pictures are unclear, which greatly reduces the readability. I suggest that the author needs to make major/substantive changes to the manuscript. See the following for specific modification details:
-Line38 “cause“-“caused“?
-Line66 “with spatial resolution“- with a spatial resolution“
-Line69 “course”-“coarse”?
-Line109 “where”-“were”?
-Figure 1(a) lacks the scale.
-Line113 Why is 2m? not 1.3m or other distance?
-Line115 What is the folder rule? Please introduce it in more detail and add corresponding references.
-Line118 Why is crown width defined as the length in the east-west direction? Why not the average distance between East and West and North and south, or the circumference or area of the crown? The latter seems more reasonable.
-Line112-119 Please add a diagram containing the method of tree height measurement and the conical crown.
-Line144 “was”-“were”?
-Line144 “manual”-“manually”?
-Line214-216 This phenomenon needs to be further explained and it is necessary to add more references.
-Line279-286 All structural variables used by authors should be referenced.
-Figures 3 and 4 should be recreated because of their low readability.
-The effect of tree shading on albedo should be mentioned and analyzed.
-Differences in tree species and vegetation types should also be analyzed.
-In the Discussion Section, the influence of factors such as slope and aspect on reflectivity is substantial. However, in order to be easier to understand and improve readability. I suggest that the author add pictures or tables for description, which can make it easier for readers to understand.
Author Response
We are grateful to the reviewer for reviewing the paper and for the constructive suggestions for improvements. Please find below the reviewer’s comments and our response to each of the comments.
REVIEWER’S COMMENT: Based on the UAV system, the author puts forward a new method to measure the albedo of Norwegian spruce, which can provide new insights into the fine reflection of the structure of Norwegian spruce. The method and experimental design of the manuscript are reasonable, and the conclusion and discussion are also substantial and reliable. However, there are still some problems that need to be paid attention to. There are a lot of spelling mistakes in the manuscript and most of the pictures are unclear, which greatly reduces the readability. I suggest that the author needs to make major/substantive changes to the manuscript.
AUTHORS’ RESPONSE: Thank you for constructive suggestions for improvements.
REVIEWER’S COMMENT: Line 38 “cause“-“caused“?
AUTHORS’ RESPONSE: Thank you. Corrected.
REVIEWER’S COMMENT: Line66 “with spatial resolution“- with a spatial resolution“
AUTHORS’ RESPONSE: Thank you. Corrected.
REVIEWER’S COMMENT: Line69 “course”-“coarse”?
AUTHORS’ RESPONSE 3: Thanks, corrected.
REVIWER’S COMMENT: Line109 “where”-“were”?
AUTHORS’ RESPONSE: Thank you. Corrected.
REVIEWER’S COMMENT: Figure 1(a) lacks the scale.
AUTHORS’ RESPONSE: Good observation. We have added the scale.
AUTHORS’ ACTION: We have redrawn Fig. 1(a) by adding the scale.
REVIEWER’S COMMENT: Line113 Why is 2m? not 1.3m or other distance?
AUTHORS’ RESPONSE: Good observation. According to visual observations and knowing that the maximum crown width was 3.38 m, a distance of 2 m was considered suitable to assure that all snow depth measurements were performed outside the zone influenced by tree crowns. In this way, the snow depth measurements were minimally influenced by snow accumulation or snow melting due to the branches. We have added a sentence explaining the choice of measurement distance.
AUTHORS’ ACTION: Improved the explanation by adding a sentence.
REVIEWER’S COMMENT: Line115 What is the folder rule? Please introduce it in more detail and add corresponding references.
AUTHORS’ RESPONSE: Thanks for clarify. Maybe this measuring tool is not widely known, however, we used nothing else than a type of height pole to measure the tree height.
AUTHORS’ ACTION: We changed “folding rule” with “height pole”.
REVIEWER’S COMMENT: Line118 Why is crown width defined as the length in the east-west direction? Why not the average distance between East and West and North and south, or the circumference or area of the crown? The latter seems more reasonable.
AUTHORS’ RESPONSE: Thank you, good observation. We decided to define the crown width in the east-west direction because this directional cross section was considered most important for the tree’s snow-masking according to the solar azimuth angle.
AUTHORS’ ACTION: We have added a sentence explaining this choice of defining the crown width.
REVIEWER’S COMMENT: Line112-119 Please add a diagram containing the method of tree height measurement and the conical crown.
AUTHORS’ RESPONSE: Good idea. Thank you.
AUTHORS’ ACTION: We have added a figure (Figure 2 in the revised manuscript) showing how tree height and snow depth were measured and calculated.
REVIEWER’S COMMENT: Line144 “was”-“were”?
AUTHORS’ RESPONSE: OK, line144 does not contain “was”. Therefore, we do not understand where this spelling mistake occur.
AUTHORS’ ACTION: No action.
REVIEWER’S COMMENT: Line144 “manual”-“manually”?
AUTHORS’ RESPONSE: OK, line 144 does not contain “manual”. We believe you mean Line211?
AUTHORS’ ACTION: We have corrected manual to manually at line 211.
REVIEWER’S COMMENT: Line214-216 This phenomenon needs to be further explained and it is necessary to add more references.
AUTHORS’ RESPONSE: We do not understand what you mean by adding more references. This phenomenon was only what we observed during the flights in field, and therefore, it is nothing to refer to. Please explain what you suggest. However, we have done a slightly modification of the first sentence (line215). Be also aware that the following sentences (line216-219), which are already written, discuss and explain this phenomenon. Maybe this clarifies what you had in mind. AUTHORS’ ACTION: We have done a slightly modification of line215.
REVIEWER’S COMMENT: Line279-286 All structural variables used by authors should be referenced.
AUTHORS’ RESPONSE: Ok, thank you. We have added a reference to the rest of the structural variables.
AUTHORS’ ACTION: Added a reference.
REVIEWER’S COMMENT: Figures 3 and 4 should be recreated because of their low readability.
AUTHORS’ RESPONSE: OK, thank you. We have recreated both figure 3 and 4. However, the figures are not unclear in the original manuscript, so this is probably happening during the process of exporting the original word document into a PDF document, which is out of our control.
AUTHORS’ ACTION: We have recreated figure 3 and 4.
REVIEWER’S COMMENT: The effect of tree shading on albedo should be mentioned and analyzed.
AUTHORS’ RESPONSE: The effect of tree shading on albedo has been addressed in terms of the snow-masking effect. This was clearly defined and derived in section 2.2. by the calculation of Ashadow. We had also done a statistical analysis of its relationship to the albedo, in addition to discuss this effect in relation to the sun’s diurnal course and interaction with the sloping terrain with different aspect. We disagree any further tree shading effect on albedo needs to be mentioned or analyzed.
AUTHORS’ ACTION. No action.
REVIEWER’S COMMENT: Differences in tree species and vegetation types should also be analyzed.
AUTHORS’ RESPONSE: This study collected and investigated only measurements of Norway spruce albedo. It is therefore impossible to analyze differences in tree species and other vegetation types based on the data material in this study. However, your suggestion is interesting, and future research should focus on how other tree species and vegetation types affect wintertime albedo in the boreal-alpine treeline.
AUTHORS’ ACTION: No action.
REVIEWER’S COMMENT: In the Discussion Section, the influence of factors such as slope and aspect on reflectivity is substantial. However, in order to be easier to understand and improve readability. I suggest that the author add pictures or tables for description, which can make it easier for readers to understand.
AUTHORS’ RESPONSE: Thank you for your suggestion. We agree that the discussion is quite massively, and that the readability was not the best. We understand your suggestion, but we do not think adding more pictures or tables will make the discussion easier to understand. This will only make the manuscript even longer without bringing any new information. The effect of slope, aspect and the diurnal course of the sun can be quite complex. However, this have been heavily reported in other studies (see references) and the readers are referred to these studies or textbooks for geometrical illustrations and explanations of the interaction of slope and aspect on albedo. Nevertheless, to improve the readability and bring attention to our main findings, we have divided the discussion into subsections with headings accordingly to the main findings of each subsection.
AUTHORS’ ACTION: We have divided the discussion into subsections in order to be easier to understand and improve readability.
Reviewer 4 Report
This study measured Boreal-Alpine Single-Tree Albedo using UAV. It is quite interesting as the authors not only presented the measurement results but also discussed some challenges. The manuscript was well prepared. I would recommend for publication on Remote Sensing.
Just one minor comment, the current manuscript is quite long. Sections 2.2-2.4 can be condensed to make it easier to read and follow.
Author Response
We are grateful to the reviewer for reviewing the paper. Please find below the reviewer’s comment and our response to it.
REVIEWER’S COMMENT: Just one minor comment, the current manuscript is quite long. Sections 2.2-2.4 can be condensed to make it easier to read and follow.
AUTHORS’ RESPONSE: Thank you. We agree the current manuscript is quite long. However, we find it difficult to condense or make sections 2.2-2.4 shorter without deteriorate the preciseness of the descriptions. It is the very first time that this methodology is described and presented. Therefore, we think the detailed descriptions presenting the complexity of the fine-spatial measurements are important. We have critically evaluated these sections, but if we remove or shorten the text, we do not consider these sections as properly described to be repeatable.
AUTHORS’ ACTION: No action.
Round 2
Reviewer 3 Report
I appreciate the authors’ work to integrate and address the suggestions from the first review. This work has greatly improved the manuscript, making it easier for the reader to follow along and understand the entire study. I have a few additional minor comments that the authors should consider addressing as they will help clarify a few items for readers.
I suggest that the author add some information about the purpose of different data obtained UAV platform before Section 2.2, which can make it easier for readers to understand.
Can be added the diurnal solar variation effect on albedo in Result 3.3 (a new paragraph)?
L555, Discussion 4.4: It would be helpful to include some discussion on the broader applications of this method(s) either to Forest Albedo.
The interaction between the diurnal course of the sun and sloping terrain constituted the most important driving factor on the albedo. But it was said separately in the discussion.
In addition, we believed that too small trees were the main reason for the weak correlation between tree structural variables and albedo in this study. Can you cite some relevant studies to illustrate the correlation between them in big trees?
Author Response
We are grateful to the reviewer for reviewing the paper and for the constructive suggestions for improvements. Please find below the reviewer’s comments and our response to each of the comments.
REVIEWER’S COMMENT: I appreciate the authors’ work to integrate and address the suggestions from the first review. This work has greatly improved the manuscript, making it easier for the reader to follow along and understand the entire study. I have a few additional minor comments that the authors should consider addressing as they will help clarify a few items for readers.
AUTHORS’ RESPONSE: Thank you! Your constructive review has been valuable to us.
REVIEWER’S COMMENT: I suggest that the author add some information about the purpose of different data obtained UAV platform before Section 2.2, which can make it easier for readers to understand.
AUTHORS’ RESPONSE: We do not understand what the intent of the comment is. Please be precise and we will act accordingly.
AUTHORS’ ACTION: No action.
REVIEWER’S COMMENT: Can be added the diurnal solar variation effect on albedo in Result 3.3 (a new paragraph)?
AUTHORS’ RESPONSE: Thank you. Since the effect of the diurnal solar variation and snow-masking are tightly related to each other, we consider it more appropriate to present these results in a joint section. This is already done in section 3.1.
AUTHORS’ ACTION: No action.
REVIEWER’S COMMENT: L555, Discussion 4.4: It would be helpful to include some discussion on the broader applications of this method(s) either to Forest Albedo.
AUTHORS’ RESPONSE: Good idea. We have inserted some sentences to explain the broader application and advantages of using fine-spatial UAV-measured albedo.
AUTHORS’ ACTION: We have inserted a few new sentences in section 4.4 in response to the reviewer’s comment.
REVIEWER’S COMMENT: The interaction between the diurnal course of the sun and sloping terrain constituted the most important driving factor on the albedo. But it was said separately in the discussion.
AUTHORS’ RESPONSE: OK, we are not quite clear on which part or which particular sentences in the discussion the comment is referring to. As described in section 4.3.1, the albedo was largely controlled by the interaction between the sloping terrain and the diurnal course of the sun. But additionally, as described in section 4.3.2, the radiation measurement error increases with increasing solar zenith angle. Naturally, the albedo is larger for large solar zenith angles, even in flat terrain. We believe we have properly explained and discussed all mechanisms driving the albedo. However, we have corrected L523: from “Since the diurnal course of the sun was the main factor driving the albedo….” to “Since the interaction between the diurnal course of the sun and the sloping terrain was the main factor driving the albedo…”. Maybe this was the sentence the reviewer found conflicting.
AUTHORS’ ACTION. We have corrected L523.
REVIEWER’S COMMENT: In addition, we believed that too small trees were the main reason for the weak correlation between tree structural variables and albedo in this study. Can you cite some relevant studies to illustrate the correlation between them in big trees?
AUTHORS’ RESPONSE: We find the comment a bit imprecise, and we are unable to find exactly what part of the text the reviewer is referring to, so please specify. We do, however, have numerous references that could be cited, see e.g., the introduction section where such references were mentioned.
AUTHORS’ ACTION: No action, but we are happy to insert one or a couple of relevant references if the reviewer can point us precisely to the text in question.